# Protocol for an implementation study of an evidence-based home cardiac rehabilitation programme for people with heart failure and their caregivers in Scotland (SCOT:REACH-HF)

Carrie Purcell [ID],[1] Paulina Daw [ID],[2] Claire Kerr,[3] J Cleland,[3] Aynsley Cowie,[4] Hasnain M Dalal [ID],[5,6] Tracy Ibbotson,[7] Clare Murphy,[8] Rod Taylor[9]

► Prepublication history and additional materials for this paper is available online. To view these files, please visit the journal online (http://dx.doi.org/10.1136/bmjopen-2020-040771).

For numbered affiliations see end of article.

**Correspondence to**
Dr Carrie Purcell;
Carrie.Purcell@glasgow.ac.uk

## ABSTRACT

**Introduction** Despite evidence that cardiac rehabilitation (CR) is an essential component of care for people with heart failure, uptake is low. A centre-based format is a known barrier, suggesting that home-based programmes might improve accessibility. The aim of SCOT: Rehabilitation EnAblement in CHronic Heart Failure (REACH-HF) is to assess the implementation of the REACH-HF home-based CR intervention in the context of the National Health Service (NHS) in Scotland.

This paper presents the design and protocol for this observational implementation study. Specific objectives of SCOT:REACH-HF are to: (1) assess service-level facilitators and barriers to the implementation of REACH-HF; (2) compare real-world patient and caregiver outcomes to those seen in a prior clinical trial; and (3) estimate the economic (health and social) impact of implementing REACH-HF in Scotland.

**Methods and analysis** The REACH-HF intervention will be delivered in partnership with four 'Beacon sites' across six NHS Scotland Health Boards, covering rural and urban areas. Health professionals from each site will be trained to facilitate delivery of the 12-week programme to 140 people with heart failure and their caregivers. Patient and caregiver outcomes will be assessed at baseline and 4-month follow-up. Assessments include the Minnesota Living with Heart Failure Questionnaire (MLHFQ), five-dimension EuroQol 5L, Hospital Anxiety and Depression Scale, and the Caregiver Burden Questionnaire. Qualitative interviews will be conducted with up to 20 health professionals involved in programme delivery (eg, cardiac nurses, physiotherapists). 65 facilitator-patient consultations will be audio recorded and assessed for fidelity. Integrative analysis will address key research questions on fidelity, context and CR participant-related outcomes. The SCOT:REACH-HF findings will inform the future potential roll-out of REACH-HF in Scotland.

**Ethics and dissemination** The study has been given ethical approval by the West of Scotland Research Ethics Service (reference 20/WS/0038, approved 25 March 2020). Written informed consent will be obtained from all participants. The study is listed on the ISRCTN registry with study ID ISRCTN53784122. The research team will ensure

### Strengths and limitations of this study

► A formal study of the implementation of a novel home-based cardiac rehabilitation programme for heart failure in the context of National Health Service Scotland.
► Study employs mixed methods which integrate quantitative and qualitative approaches to understand the implementation process.
► Addresses home-based cardiac rehabilitation at a time of increased interest in, and need for, remote facilitation of care due to the COVID-19 pandemic.
► Although limited to four sites geographical sites, these sites incorporate a wide range of settings including urban and rural populations.

that the study is conducted in accordance with both General Data Protection Regulations and the University of Glasgow's Research Governance Framework. Findings will be reported to the funder and shared with Beacon Sites, to facilitate service evaluation, planning and good practice. To broaden interest in, and understanding of REACH-HF, we will seek to publish in peer-reviewed scientific journals and present at stakeholder events, national and international conferences.

## INTRODUCTION

Heart failure (HF) is both serious and common, and its prevalence is increasing.[1 2] Despite advances in care, people in Scotland with HF continue to have worse survival rates than those of some common cancers.[3] HF often has a negative effect on health-related quality of life (HRQoL) for those living with it,[1 4] and carries a high risk of hospitalisation, a major driver of the economic burden.[1 5]

Cardiac rehabilitation (CR) is highly effective, cost-effective and integral to comprehensive care of people with HF.[6–8] Self-care in HF is also widely acknowledged

as important, and should also involve family/friends, and promote self-efficacy.[9] A recent individual–participant data meta-analysis,[7] and updated Cochrane review, show that, compared with no rehabilitation, CR participation reduces the risk of all-cause hospitalisation and improves HRQoL (assessed using the MLHFQ).[8] The 2019 National Heart Failure Audit reported that referral for CR was associated with a 12% reduction in mortality.[10]

Despite this strong evidence, and national and international clinical guidelines recommending that anyone living with HF should receive CR, referral for and participation in CR remains low.[8] The National Audit of Cardiac Rehabilitation (NACR) found that only 57% of people with HF in England, Wales and Northern Ireland (Scottish data are not currently included in this audit) who were offered CR in 2018–2019 attended one or more sessions (email communication from NACR). Currently, most cardiac patients (77%) receive centre (hospital)-based, group CR.[8] Travelling to centres and dislike of group exercise are key barriers to participation in centre-based programmes.[6 10 11] That women, people from black and minority ethnic groups, and those living in high deprivation are less likely to attend centre-based CR,[8] indicates that centre-based approaches are exclusionary. Home-based CR thus offers a cost-effective approach to improving CR uptake by people with HF, resulting in better health and well-being outcomes. The 2020 COVID-19 pandemic, and the policy by many countries of home lockdown to maintain social distancing, has dramatically underlined the urgent need for alternatives to centre-based models of healthcare provision.

We codeveloped (with clinicians/practitioners, people with health failure, their caregivers and service commissioners) an evidence-based and theory-based, novel home CR intervention for people with HF: Rehabilitation EnAblement in CHronic Heart Failure (REACH-HF).[12] A multicentre randomised trial demonstrated that the addition of REACH-HF to usual medical care resulted in a clinically important improvement in HRQoL of people with HF, when compared with usual care alone.[13] Economic modelling showed that the REACH-HF intervention to be both low-cost (at £417/patient) and cost effective.[14] However, there remains a paucity of data regarding the extent to which introducing home-based CR for HF increases CR uptake.[6 15] Moreover, it is uncertain that the positive outcomes identified in the REACH-HF randomised controlled trial (RCT) can be replicated in a 'real-world' setting, and what key considerations are with regards to embedding such an intervention in everyday practice.

At present, relatively few evidence-based healthcare interventions become embedded in routine clinical practice.[16] Factors contributing to this include: weak external validity of efficacy trials; intervention developers' lack of consideration for scale-up; trial design issues; and development of interventions that are overly theoretical.[17–19] Where implemented, evaluations often consider individual-level health professional performance,

targeting knowledge, routines and attitudes.[20 21] Individuals play a significant role in implementation, in that they dynamically engage with interventions while, to varying degrees, embodying their own interests and motivations and those of their profession, organisation and culture.[22 23] It is crucial also to understand community, organisational, system and policy-level influences on the embedding of innovative practice.[22]

Running parallel to a similar implementation study already underway in England and Northern Ireland,[24] our study seeks to understand the organisational and other wide-ranging influences affecting the implementation of REACH-HF in Scotland, in order to inform potential large-scale roll-out of the intervention. A key factor shaping implementation is that a given intervention may not produce the same effects when transferred from one context to another and, crucially, from a randomised trial to the real world. Target population characteristics may differ in key ways, such as geographical location (urban/rural) or relative deprivation. Moreover, there may be significant contextual differences between sites and teams delivering a healthcare intervention, such as the size of the team or familiarity with a given approach. Such contextual differences may produce adaptations in what is delivered and how (ie, impacting fidelity to the intervention design). This may in turn shape intervention results—including any proven benefit—when compared with an RCT.[25] Adaptability to context may also impact the sustainability of an intervention, that is, the extent to which it is embedded in everyday practice.[22]

We draw specifically from UK Medical Research Council (MRC) guidance on evaluation of complex interventions, particularly using process evaluation methodology.[26] Process evaluation is an established means by which to understand implementation by assessing: fidelity (the degree to which the intervention was delivered as intended); context (barriers to and facilitators of implementation, including those that might explain variation in outcomes), and mechanisms of impact.[23] As the mechanisms by which the REACH-HF intervention changes behaviour have been described and explored elsewhere,[12 13] we focus here on fidelity and context in the new delivery setting. Integration of process and outcome data can generate better understanding of, for example, whether and how adaptations to implementation, or differences among sites, explains any observed variation in outcomes, as well as informing improvements for future roll-out.

## METHODS AND ANALYSIS
### Study design
A mixed-method implementation study comprises a multicentre prospective cohort study and nested process and economic evaluations.

The overarching aim of this study is to assess the real-world implementation REACH-HF for people living with

HF and their caregivers in Scotland. Our research questions are:

1. What are the service-level facilitators and barriers to the implementation of REACH-HF?
2. How do real-world patient and caregiver outcomes compare with those seen in a prior clinical trial?
3. What is the estimated economic (health and social) impact of implementing REACH-HF in Scotland?

Informed by process evaluation methodology, the study protocol detailed below is thus organised around four key components, which contribute to answering these questions:

► Fidelity of implementation: what was implemented and how closely this reflected what was intended (ie, the original REACH-HF intervention) (RQ1&2).
► Contextual factors: barriers to, and facilitators of, implementation, as perceived by the health professionals and service organisers involved; 'background noise' to implementation (RQ1).
► CR participant-related outcomes: whether, and to what extent, improvements in patient outcomes seen in the REACH-HF RCT are replicated (RQ2).
► Economic impact: health and social implementation costs (RQ3).

The study will be conducted across Scottish NHS Health Board CR services which, as early adopters of REACH-HF, will be designated as 'Beacon Sites'. (The use of early adopters to model intervention implementation is itself one means of contributing to routinisation/embedding of innovative practice.[22]) A national application process followed promotion at national conferences, and contact letters to HF specialist nurses and CR leads. This resulted in recruitment of four sites across six NHS Health Boards to act as Beacon Sites: NHS Ayrshire and Arran; NHS Lanarkshire; NHS Forth Valley; and NHS Highland, Orkney, and Shetland (combined to act as one site due to small patient numbers). Sites were selected for their ability to commit to delivery of REACH-HF, and for geographic spread.

We will assess patient outcomes before and after administering the 12-week programme with 35 people with HF (140 total). Members of the HF team at each site will be interviewed. Detailed information of the costs and utilisation of the provision of the REACH-HF programme will be collected. Given the ongoing COVID-19 crisis, the start of data collection for the study has been delayed, but will begin in November 2020. NHS Greater Glasgow and Clyde will act as study sponsor.

## Sample and recruitment

The study will be conducted across Scottish NHS 'Beacon Sites'. People with HF are eligible if they: are aged 18 years or over; have a confirmed diagnosis of systolic (reduced ejection fraction) HF within the past 5 years; have experienced no deterioration of HF symptoms in the preceding 2 weeks resulting in hospitalisation or alteration of HF medication; and are deemed suitable for CR by their local clinical team. We will exclude anyone who: has undertaken CR in the preceding 12 months; has medical contraindications to exercise testing or training; is in a long-term care establishment, or unwilling/unable to travel to research assessments or accommodate home visits; is unable to understand the study information or unable to complete the outcome questionnaires. Patients with a caregiver will also be invited to participate. Patients with no caregiver, or whose caregivers do not wish to participate, are still eligible take part in the study.

Sites will recruit people with HF, using their usual means of CR referral to introduce the study. This is likely to include a variety of pathways such as: people with HF referred for CR from acute or primary care; review of patients held on site HF databases; and approaching people with HF at outpatient appointments/home visits. Potential participants will be provided with invitation letters, information sheets, and reply slips for both them and their caregiver (if applicable), and those interested in participation will be asked to instigate contact with the research team by returning the reply slips. Figure 1 outlines the participation pathway for people with HF.

A maximum of 20 individual interviews will be conducted with health professionals involved in the delivery of SCOT:REACH-HF, near the end of the

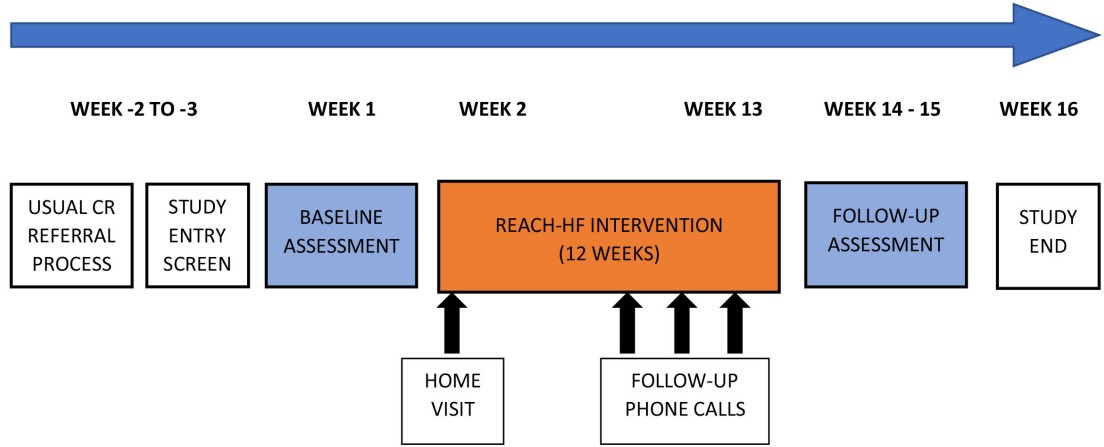

**Figure 1** SCOT: Rehabilitation EnAblement in CHronic Heart Failure (REACH-HF) participant pathway.

intervention period. These will include the trained facilitators (typically CR nurses or physiotherapists), as well as other key individuals involved in coordinating and commissioning CR (such as senior clinicians and service management). There may be some variation in participant roles due to the differing structures of local HF teams. We will use a combination of convenience and purposive sampling, offering the opportunity to participate to all those delivering REACH-HF, and those identified as having a key role in service planning, to ensure capture of diverse perspectives. A participant information sheet will be provided to all potential interviewees, and they will have adequate time to consider participation and ask questions about the interview process. Written consent will be obtained prior to face-to-face interviews. Where interviews are to be conducted by telephone, consent forms will be completed digitally and returned by email, and verbal consent recorded digitally.

## The intervention

REACH-HF is a home-based, health professional-facilitated, 12-week CR programme supporting self-care in patients with HF. Three health professionals with CR and/or HF experience from each Beacon Site will attend a 2 days online REACH-HF training course facilitated by the Heart Manual Department, NHS Lothian (formerly a 3 days face-to-face training delivered in Edinburgh, the course has been adapted to accommodate current restrictions). Training focuses on the seven steps of successful facilitation of REACH-HF (in turn based on a person-centred care approach): (1) build rapport; (2) assess needs and build understanding of HF; (3) support self-management and progress monitoring; (4) discuss exercise and well-being; (5) summarise and plan next steps; (6) review progress; (7) support long-term maintenance. As such, training includes sessions on psychology, behaviour change, physical activity, engaging caregivers and newly adapted components to address intervention delivery during the COVID-19 pandemic. Following training, facilitators are then asked to implement REACH-HF. The programme—described in detail elsewhere[11][12]—is outlined in box 1.

## Measures/data collection

Patient and caregiver outcome data will be collected during an initial assessment appointment by a designated member of the Beacon Site team trained in data collection by the REACH-HF team, and via self-completion questionnaires (either postal or online, as per participants' preference). Data will be collected at baseline—before commencing with the REACH-HF programme—and 4 months following baseline, which coincides with the end of intervention delivery period (see figure 1).

## Fidelity assessment

Facilitator–patient interactions (face-to-face and/or phone) for 65 participants will be audio recorded. Recordings will be assessed using our established fidelity

---

**Box 1 The Rehabilitation EnAblement in CHronic Heart Failure (REACH-HF) Intervention**

► *The Heart Failure Manual*, which provides information about HF for the person with HF, to increase understanding of the condition and address common misconceptions.
► Information on and strategies for managing HF, and further relevant advice on, for example, managing lifestyle risk, managing depression and anxiety, and getting support from others.
► A choice of two exercise training programmes: a chair-based programme (via DVD and online) and a walking programme; with a recommendation that these should be engaged in three times weekly, alongside general physical activity.
► A stress-management programme, with relaxation techniques (provided in the manual and in audio format) to help cope with anxiety and depression.
► A progress tracker designed to facilitate an individual's learning from experience through self-monitoring of behaviour and symptoms. (this prompts help seeking as appropriate).
► A *Family and Friends Resource* to increase caregiver understanding of HF, to enable them to support the person with HF's self-care and well-being.
► Face-to-face and telephone facilitation over 12 weeks by a health professional trained to deliver the REACH-HF programme.

---

assessment checklist (described in detail elsewhere[12]). This 12-item checklist focuses on assessing inclusion by facilitators of key processes such as patient-centred communication, making a plan of action, and encouraging self-monitoring of progress. Facilitators will also be asked to complete a brief self-rated fidelity checklist after every session. This comprises questions on the same 12 programme components and asks facilitators to rate occurrences of each feature (absent, minimal, some, sufficient, good, very good, excellent). An independent observer rating is resource intensive, while self-rated assessment may provide a pragmatic, real-world alternative to monitor delivery quality. We will also explore (in the interviews below) whether use of the checklist facilitates/encourages reflexive practice and, in doing so, quality of implementation.

## Context

We seek to capture data on barriers to and facilitators of implementation REACH-HF by interviewing health professionals at each Beacon Site. We anticipate conducting up to 20 individual interviews, which will be audio recorded and transcribed verbatim for analysis. Interviews will be conducted by CP face-to-face or by phone, as per the participant's preference. Normalisation Process Theory (NPT)[27] will be used as a theoretical framework to guide data collection, analysis and interpretation. A flexible topic guide—informed by the four constituent constructs of NPT (coherence, cognitive participation, collective action, reflexive monitoring), the existing literature on CR, and the key aims of the implementation study—will facilitate generation of rich data,

as well as enabling capture of factors unanticipated by the research team (see online supplemental appendix 1).

Additional ad hoc contextual data from each site will be collated centrally (by CP) in one implementation log (Excel file) which will also capture overall 'background noise' to implementation (such as the impact of the COVID-19 pandemic) which will contribute to the contextual analysis.

### CR participant-related outcomes

Data will be recorded in an electronic case report form (CRF), and participants will be offered the option of a paper self-completion questionnaire or a secure individual link sent by email to complete the questionnaire online. At the baseline appointment, after obtaining written consent, Beacon Site teams will collect medical history from the participants' hospital and primary care records, including: comorbidities (number and severity scored with Charlson Comorbidity Index); New York Heart Association class; HF aetiology; concomitant HF medication and presence of implantable HF devices.

Participants will provide detailed sociodemographic data (age, gender, ethnicity, weight, employment status, education level, smoking status) at baseline. The following participant outcomes will be assessed: disease-specific HRQoL measured using the MLHFQ; generic quality of life (five-dimension EuroQol (EQ-5D-5L) scale); psychological well-being (Hospital Anxiety and Depression Scale (HADS)); patient-reported outcome measure for cardiac rehab; hospitalisations and primary care contacts (number, reason, duration); adverse events (eg, skeletomuscular injury); health literacy (Health Literacy Questionnaire); and, if possible, exercise capacity via an incremental shuttle walk test (if face-to-face assessment possible). Caregiver outcomes are: generic quality of life (EQ-5D-5L); psychological well-being (HADS); Family Caregiver Quality of Life Scale questionnaire; Caregiver Burden Questionnaire HF; Caregiver Contribution to Self-care of HF Index questionnaire. The same outcomes will be collected at the 4-month follow-up.

### Economic impact

Data will be collected to allow the costing of the REACH-HF intervention delivery. These will include time input from REACH-HF facilitators, supervision for facilitators, training costs for facilitators and consumables. Unit costs for resource use will be sought from national published or NHS sources. Data from each site will be recorded in the implementation log (excel file, as above).

Additionally, facilitators will be asked to complete a Facilitator Log for each participant. This log is a one-page pro forma designed to capture time, expenditure and any other resources required for the implementation of REACH-HF, as well as any adaptations made to the intervention for individual patients. As such, it will capture essential data for the fidelity and economic analyses. Completed forms will be returned to the research team for data entry and analysis.

### Data management

Data management will follow the principles of Good Clinical Practice and supported by the University of Glasgow (UoG) Clinical Trials Unit (GCTU). An electronic CRF (eCRF) developed by the GCTU will capture all data noted above. Access to the eCRF will be restricted, via a study-specific web portal, with only authorised personnel able to make entries. RT or their designee will be responsible for all eCRF entries, and will confirm that data are accurate, complete and verifiable. Entries from participant medical records will be made locally by Beacon Site staff trained by the research team. Where data are entered by the participant into a paper CRF, completed anonymous questionnaires will be returned by post to the University of Glasgow for data entry. Where completed by the participant electronically, data will be entered directly into a participant-facing version of the eCRF. Where practical, data will be validated at the point of entry into the eCRF. Any additional data discrepancies will be flagged to RT and any changes recorded to maintain a complete audit trail (reason, date and who made the change). Data will be stored in a MS SQL Server database. Direct access to the study web portal will be granted, on request, to authorised representatives of the sponsor, host institution and regulatory authorities to permit trial-related monitoring, audits and inspections.

The qualitative, fidelity and economic impact components of the study will be conducted by UoG under the direction of RT. Transcription will be undertaken by a specialist service with whom UoG has an ongoing contractual arrangement and confidentiality agreement. All data (Excel files, audio recordings and anonymised transcripts, stored separately) will be kept for at least ten years in line with UoG Research Governance Framework Regulations for clinical research. Data will be stored confidentially on password-protected servers maintained on the UoG network. Anonymised data will be made available to other legitimate researchers on request, as per study consents.

The study will be overseen by the Project Management Group (coapplicants) and Project Advisory Group (national CR experts)—see online supplemental appendix 2 for membership.

### Data analysis

We require 130 participants to detect pre–post intervention change in the MLFHQ scores to achieve the minimal important difference[13] ≥5 points. This calculation is based on a MLHFQ SD of 24 points, within patient pre–post correlation (r=0.72) and attrition rate of ≤10% as seen in our multicentre RCTs refs. There is no formal sample size calculation for the number of caregivers participating in this study.

### Fidelity

Fidelity data will be analysed by CP. Fidelity checklist scores will be collated at facilitator, site and total sample levels. We will present descriptive statistics (means, ranges), using the same analytic approach as the original

REACH-HF trial.[13] In brief, the fidelity checklist uses an established 0–6 scale (Dreyfus scale of clinical skills acquisition[28]) to rate clinical skills, and is anchored such that a score of three or more represents adequate delivery quality for each item. Fidelity outcomes will be compared with the REACH-HF RCT,[13] and analysed alongside self-rated fidelity scores. Overall findings will be integrated with the context and CR participant-related outcome data findings.

### CR participant-related outcomes

The primary analyses for primary and secondary quantitative outcomes will based on a within-patient comparison in participants with complete outcome data at 4 months. We will examine the characteristics of any patients who withdraw, and conduct secondary analysis based on imputation of their missing outcome data. All within-patient outcome comparisons will be presented as mean difference with 95% CI. The outcome effect size seen in the Beacon Sites will be indirectly compared with the changes found in the REACH-HF trial.[13] Statistical analysis will be conducted by RT using STATA V.15. Descriptive statistics will be presented in order to describe study population characteristics.

### Context

Verbatim transcripts (Word documents) will be pseudonymised (removing any potential indicators of personal identity or site) and uploaded into NVivo V.12 qualitative software to facilitate data management. A coding framework will be developed, informed by the constructs of NPT noted above, and taking an approach informed by the Framework method.[29] This approach will also for consideration of unanticipated issues.

Following this categorising stage, a further interpretive stage will see data examined across sources (professional role) and cases (sites). This will facilitate understanding of contextual factors shaping implementation of REACH-HF in context, and development of potential explanations for commonalities and differences between our findings and the previous RCT.

A subsequent integrative analysis will be conducted to bring together each analytic component (fidelity, context, CR participant-related outcomes). Integrative analysis will involve placing all relevant data in one integrative matrix and assessing for synergies which indicate our key findings, again guided by the NPT framework. Placing key findings in a matrix alongside those from the original REACH-HF RCT will also facilitate understanding of the real-world effectiveness of the intervention. First stage coding, interpretation will be conducted by CP in consultation with RT. Integrative analysis will be conducted by CP and RT with input from the project management group.

### Economic impact

Economic analysis will focus on assessing the cost of the delivery of REACH-HF in the four Beacon Sites, that is, the additional (incremental) costs associated with delivery of the HF Manual, when added to usual care. Healthcare costs will be estimated using resource use data collected within the study, and unit costs for resource use from national published/NHS sources. Resource use is expected to consist of time input from REACH-HF facilitators, supervision for facilitators, training costs for facilitators and consumables (eg, intervention booklets for participants and facilitators). Data on facilitator time will be captured by facilitators at participant level, using the Facilitator Log described above. Economic analysis will be conducted by CP and RT alongside the main statistical analysis.

### Patient public involvement (PPI)

People with HF and their caregivers had an extensive input into the development of the REACH-HF intervention, and a substantial body of data on patient experiences has been generated through interviews with RCT participants.[13 14] We have established a standing PPI group for SCOT:REACH-HF led by TI, involving people with HF and their caregivers, who are independent of the study. Four meetings of the PPI group will be convened during the study to review participant-facing materials, advise on dissemination, and provide input on any participant related problems that may arise, such as recruitment and retention.

## DISCUSSION

Approaches to implementation science are varied.[23] We draw on MRC guidance on the evaluation of complex interventions which highlights that, while RCTs are viewed by many as the gold standard for demonstrating efficacy, they do not tell policy-makers or service commissioners whether an intervention would produce the same outcomes in their context.[26] A process evaluation approach produces understanding of implementation by assessing fidelity, context, and mechanisms of impact.[25] As the mechanisms of REACH-HF are explored elsewhere,[12 13] this study focuses on fidelity and context in the new delivery setting.

Most complex interventions would be expected to see some adaptation as they are transferred into real-world settings[23] (variable by how much contextual factors have been considered in the design process). Indeed, some adaptability is in fact desirable in order to support effectiveness.[30] In order to assess if and how any adaptations might have impacted the overall integrity of the intervention, it is vital to (1) have a clear picture ahead of implementation of what the active components of an intervention are, and (2) understand how closely delivery follows what is intended.[30] Hence, we include above a description of the intervention's constituent parts, and include in the study design a multipronged approach to assessing fidelity.

There are limitations on the degree to which novel interventions become embedded in routine clinical

practice. However, these limitations can be ameliorated by well-considered studies of implementation. By operationalising a tailored process evaluation methodology, we aim to assess such implementation, and the translation from RCT to real world, by paying particular attention to: fidelity to the intended programme, contextual factors shaping delivery, and how these may explain any differences measured in participant outcomes.

Our findings will inform potential larger scale roll-out of REACH-HF, offer guidance to policy-makers and CR commissioners, inform contextual adaptations, and facilitate diffusion and embedding of home-based CR for people with HF in the UK.

## Strengths and limitations

This study will formally assess of the implementation of a novel home-based CR programme for HF in the context of NHS Scotland. It employs mixed methods which integrates quantitative and qualitative approaches to understanding the implementation process. Moreover, our study will facilitate a communication channel between researchers and implementers, in order to support high-quality services for people with HF, and establish four key Beacon Sites that have the potential to model intervention roll-out, should that be adopted more widely. Although limited to four sites geographical sites, these sites incorporate a wide range of settings including urban and rural populations. An additional strength is the adaptation of the study to the restrictions of the ongoing COVID-19 pandemic, and the potential to assess implementation of support for self-care for a potentially vulnerable population.

## ETHICS AND DISSEMINATION

The study has been given ethical approval by the West of Scotland Research Ethics Service (reference 20/WS/0038, approved 25 March 2020). Written informed consent will be obtained from all participants. The study is listed on the ISRCTN registry with study ID ISRCTN53784122. The research team will ensure that the study is conducted in accordance with both General Data Protection Regulations and the University of Glasgow's Research Governance Framework. Findings will be reported to the funder and shared with Beacon Sites, to facilitate service evaluation, planning and good practice. To broaden interest in, and understanding of REACH-HF, we will seek to publish in peer-reviewed scientific journals and present at stakeholder events, national and international conferences.

## Author affiliations
[1]MRC/CSO SPHSU, University of Glasgow, Glasgow, UK
[2]School of Sport, Exercise & Rehabilitation Sciences, University of Birmingham, Birmingham, UK
[3]Robertson Centre for Biostatistics, University of Glasgow, Glasgow, UK
[4]Cardiac Rehabilitation, University Hospital Crosshouse, NHS Ayrshire and Arran, Kilmarnock, UK
[5]Royal Cornwall Hospitals NHS Trust, Truro, UK
[6]College of Medicine and Health, University of Exeter Medical School, Exeter, UK
[7]Institute of Health and Wellbeing, University of Glasgow, Glasgow, UK
[8]Royal Alexandra Hospital, NHS Greater Glasgow and Clyde, Glasgow, UK
[9]MRC/CSO Social and Public Health Sciences Unit and Robertson Centre for Biostatistics, University of Glasgow, Glasgow, UK

**Contributors** The study was conceived of and designed by RT, HD and the REACH-HF collaboration. CP and RT drafted the manuscript. RT and CP will lead recruitment and set-up of Beacon Sites, and CP will oversee day-to-day project management. RT, CK and CP secured all relevant ethical approvals for the project and prepared all study documentation, to which PD also contributed. AC designed the PROM-CR measure. CP and RT will conduct study analysis and write-up. TI will lead on coordinating PPI. JC and CM will provide project supervision and oversight. All authors reviewed and approved the final version of this protocol for publication.

**Funding** This work is supported by Heart Research UK (grant SC09/19) and conducted by the MRC/CSO Social and Public Health Sciences Unit, University of Glasgow (grants SPHSU14 and MC_UU_12017/14). RT is part funded by the National Institute for Health Research (NIHR) Programme Grants for Applied Research scheme (project reference RP-PG-1210-12004).

**Competing interests** None declared.

**Patient consent for publication** Not required.

**Provenance and peer review** Not commissioned; externally peer reviewed.

**ORCID iDs**
Carrie Purcell http://orcid.org/0000-0002-2651-9201
Paulina Daw http://orcid.org/0000-0002-0942-3953
Hasnain M Dalal http://orcid.org/0000-0002-7316-7544

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
