## [Reviewer comments · BMJ Open]

ARTICLE DETAILS

TITLE (PROVISIONAL)	Protocol for an implementation study of an evidence-based home cardiac rehabilitation programme for people with heart failure and their caregivers in Scotland (SCOT:REACH-HF)
AUTHORS	Purcell, Carrie; Daw, Paulina; Kerr, Claire; Cleland, J; Cowie, Aynsley; Dalal, Hasnain; Ibbotson, Tracy; Murphy, Clare; Taylor, Rod

VERSION 1 – REVIEW

REVIEWER	David E Winchester Malcom Randall VAMC, USA University of Florida College of Medicine, USA
REVIEW RETURNED	30-Jun-2020

GENERAL COMMENTS	This manuscript describes the design of an observational study intended to increase adoption of cardiac rehabilitation in Scotland. Methods being used include validated instruments related to quality of care for chronic CV conditions and qualitative interviews with clinicians involved in the program. Introduction: Clearly elucidates the reasoning behind the study and relevant background on center-based and home-based cardiac rehabilitation programs. Methods: The authors propose to gather data from CR participants with several instruments, protocol would be strengthened if the authors could describe how these data will be compared to prior RCT data to compare real-world effectiveness (as described in RQ2) Participants will provide feedback on the program through the PPI group, have the authors considered adding interviews with patients to complement those of the clinicians? Is an Implementation Science framework being applied to the entire study, or just the qualitative interview guide? Who will be responsible at each site for completing the implementation log? Will any monitoring be done to ensure this is done throughout the study? It sounds as if the interview instrument has not been developed yet, if possible it would make a strong addition to the protocol manuscript.
---

REVIEWER	Jessica Orchard University of Sydney / HRI
REVIEW RETURNED	03-Jul-2020

GENERAL COMMENTS	Thank you to the authors for designing this important implementation study of home-based CR. The protocol is very well written and clear. I have a couple of very minor suggestions:  1. Some additional detail about the economic analysis proposed would be helpful 2. Has the trial been registered? If so, please include the details 3. If possible, it would be interesting to also obtain follow-up data longer after the intervention has finished (eg 6-12 months) to see whether results have been sustained. Thank you and good luck with the study.
---

VERSION 1 – AUTHOR RESPONSE

Reviewer: 1

Reviewer Name: David E Winchester

Institution and Country:

Malcom Randall VAMC, USA

University of Florida College of Medicine, USA

Competing interests: None declared

Please leave your comments for the authors below

This manuscript describes the design of an observational study intended to increase adoption of cardiac rehabilitation in Scotland. Methods being used include validated instruments related to quality of care for chronic CV conditions and qualitative interviews with clinicians involved in the program.

Introduction:

Clearly elucidates the reasoning behind the study and relevant background on center-based and home-based cardiac rehabilitation programs.

Methods:

The authors propose to gather data from CR participants with several instruments, protocol would be strengthened if the authors could describe how these data will be compared to prior RCT data to compare real-world effectiveness (as described in RQ2)

Note added on p12 to indicate that this will form part of the integrative analysis: 'Placing key findings in a matrix alongside those from the original REACH-HF RCT will also facilitate understanding of the 'real world' effectiveness of the intervention' and that 'The outcome effect size seen in the Beacon Sites will be indirectly compared to the changes found in the REACH-HF trial'.

Participants will provide feedback on the program through the PPI group, have the authors considered adding interviews with patients to complement those of the clinicians?

The PPI group participants are independent of the study (clarification added on p12). Extensive data has been generated on the experiences of people with heart failure and their caregivers, in the process evaluation of the REACH-HF study (note added to p12). We thus took the decision not to incorporate further interviews with participants in the design of this study, but to focus instead on service-level barriers and facilitators.

Is an Implementation Science framework being applied to the entire study, or just the qualitative interview guide?

The NPT framework will guide the overall integrative analysis as well as the interviews – clarification added on p12.

Who will be responsible at each site for completing the implementation log? Will any monitoring be done to ensure this is done throughout the study?

The implementation data will be collated centrally by CP (note added on p9). However, facilitators will also be asked to complete a facilitator log following each session – as this was not previously noted in the manuscript, this information has now been added to p9.

It sounds as if the interview instrument has not been developed yet, if possible it would make a strong addition to the protocol manuscript.

The topic guide has now been included as Appendix 1.

Reviewer: 2

Reviewer Name: Jessica Orchard

Institution and Country: University of Sydney / HRI

Competing interests: None declared

Please leave your comments for the authors below

Thank you to the authors for designing this important implementation study of home-based CR. The protocol is very well written and clear.

I have a couple of very minor suggestions:

1. Some additional detail about the economic analysis proposed would be helpful

Text now added on p.12-13 to elaborate on the planned economic analysis.

2. Has the trial been registered? If so, please include the details

The trial registration is now included in the Abstract.

3. If possible, it would be interesting to also obtain follow-up data longer after the intervention has finished (eg 6-12 months) to see whether results have been sustained.

We agree that a more longitudinal approach would enable us to establish the extent to which the intervention has become embedded as routine practice. This has not been possible within the constraints of the current study funding, but we will actively explore the opportunity for such a study extension of data collection.

Thank you and good luck with the study.

VERSION 2 – REVIEW

REVIEWER	Jessica Orchard University of Sydney / HRI
REVIEW RETURNED	19-Aug-2020
GENERAL COMMENTS	Thanks for the revised manuscript and good luck with the study.